# Development and validation of a predictive model of in-hospital mortality in COVID-19 patients

Diego Velasco-Rodríguez[1], Juan-Manuel Alonso-Dominguez[1]*, Rosa Vidal Laso[1], Daniel Lainez-González[1], Aránzazu García-Raso[1], Sara Martín-Herrero[1], Antonio Herrero[2], Inés Martínez Alfonzo[1], Juana Serrano-López[1], Elena Jiménez-Barral[1], Sara Nistal[3], Manuel Pérez Márquez[4], Elham Askari[1], Jorge Castillo Álvarez[5], Antonio Núñez[6], Ángel Jiménez Rodríguez[7], Sarah Heili-Frades[8], César Pérez-Calvo[4], Miguel Górgolas[5], Raquel Barba[3], Pilar Llamas-Sillero[1]

1 Department of Hematology, Hospital Universitario Fundación Jiménez Díaz, IIS-FJD, Madrid, Spain, 2 Department of Information Technology, Quironsalud, Madrid, Spain, 3 Department of Internal Medicine, Hospital Universitario Rey Juan Carlos, Móstoles, Madrid, Spain, 4 Intensive Care Unit, Hospital Universitario Fundación Jiménez Díaz, IIS-FJD, Madrid, Spain, 5 Department of Internal Medicine, Hospital Universitario Fundación Jiménez Díaz, IIS-FJD, Madrid, Spain, 6 Department of Internal Medicine, Hospital General de Villalba, Collado Villalba, Madrid, Spain, 7 Department of Internal Medicine, Hospital Infanta Elena, Valdemoro, Madrid, Spain, 8 Department of Pneumology, Hospital Universitario Fundación Jiménez Díaz, IIS-FJD, Madrid, Spain

* juan.adominguez@fjd.es

**Data Availability Statement:** All data will be uploaded as Supporting Information file.

## Abstract

We retrospectively evaluated 2879 hospitalized COVID-19 patients from four hospitals to evaluate the ability of demographic data, medical history, and on-admission laboratory parameters to predict in-hospital mortality. Association of previously published risk factors (age, gender, arterial hypertension, diabetes mellitus, smoking habit, obesity, renal failure, cardiovascular/ pulmonary diseases, serum ferritin, lymphocyte count, APTT, PT, fibrinogen, D-dimer, and platelet count) with death was tested by a multivariate logistic regression, and a predictive model was created, with further validation in an independent sample. A total of 2070 hospitalized COVID-19 patients were finally included in the multivariable analysis. Age 61–70 years ($p$<0.001; OR: 7.69; 95%CI: 2.93 to 20.14), age 71–80 years ($p$<0.001; OR: 14.99; 95%CI: 5.88 to 38.22), age >80 years ($p$<0.001; OR: 36.78; 95%CI: 14.42 to 93.85), male gender ($p$<0.001; OR: 1.84; 95%CI: 1.31 to 2.58), D-dimer levels >2 ULN ($p$ = 0.003; OR: 1.79; 95%CI: 1.22 to 2.62), and prolonged PT ($p$<0.001; OR: 2.18; 95%CI: 1.49 to 3.18) were independently associated with increased in-hospital mortality. A predictive model performed with these parameters showed an AUC of 0.81 in the development cohort (n = 1270) [sensitivity of 95.83%, specificity of 41.46%, negative predictive value of 98.01%, and positive predictive value of 24.85%]. These results were then validated in an independent data sample (n = 800). Our predictive model of in-hospital mortality of COVID-19 patients has been developed, calibrated and validated. The model (MRS-COVID) included age, male gender, and on-admission coagulopathy markers as positively correlated factors with fatal outcome.

**Funding:** The authors received no specific funding for this work. Quironsalud provided support in the form of salaries for all authors, but did not have any additional role in the study design, data collection and analysis, decision to publish, or preparation of the manuscript. The specific roles of these authors are articulated in the 'author contributions' section.

**Competing interests:** All the authors are employees of Quironsalud. This does not alter our adherence to PLOS ONE policies on sharing data and materials.

# Introduction

Initial symptoms of the disease produced by SARS-CoV-2, 2019-nCoV (COVID-19), are similar to other viral syndromes, but COVID-19 has the potential to develop a systemic inflammatory response syndrome, acute respiratory distress syndrome (ARDS), multi-organ failure and shock, especially in older patients with comorbidities [1–4]. The COVID-19 pandemic has spread to the whole world [2], causing over 1.5 million deaths to date.

Several factors have been correlated with higher mortality in these patients: older age [5–7], male gender [6, 8], arterial hypertension [8], diabetes [5, 8], smoking [8], obesity [7], cardiac and pulmonary pathology [5, 8], and lymphopenia [9].

Several studies have described that severe COVID-19 disease is frequently complicated with coagulopathy [10–13]. However, COVID-19 associated coagulopathy behaves predominantly as a pro-thrombotic status rather than a bleeding disorder [11, 14]. High fibrinogen levels and normal or slightly low platelet counts are usually found [11, 14], unlike "classical" overt DIC [15]. These patients show not only high venous thromboembolism (VTE) rates, up to 16–27%, even despite having received adequate VTE prophylaxis [16, 17], but also cardiovascular complications [18]. Pathological evidence of pulmonary microthrombosis in severe cases has also been provided [19]. Among coagulation parameters, elevated D-dimer levels show a strong correlation with mortality [3, 13, 20].

Early and effective predictive models of clinical outcomes are necessary for risk stratification of hospitalized COVID-19 patients, especially if there is a high volume of patients consulting in the emergency departments [11]. Clinicians need better predictors of mortality and tools capable to detect which patients are prone to deteriorate rapidly. Our aim was to evaluate the ability of demographic data, medical history, and on-admission laboratory parameters to predict mortality in hospitalized COVID-19 patients.

# Materials and methods

## Patients and sample handling

Two thousand eight hundred and seventy nine consecutive hospitalized adult patients with confirmed moderate or severe COVID-19 from four hospitals [Hospital General de Villalba (Collado Villalba, Madrid), Hospital Infanta Elena (Valdemoro, Madrid), Hospital Universitario Rey Juan Carlos (Móstoles, Madrid) and Hospital Universitario Fundación Jiménez Díaz in Madrid] from February 27 to April 17, 2020, were retrospectively evaluated. COVID-19 was considered at least moderate and required hospitalization if any of these criteria was met: CURB-65 score >2 or FINE>II, peripheral capillary oxygen saturation (SpO$_2$) <93% or respiratory rate >20 breaths per minute or PaO$_2$ <65 mmHg, bilateral infiltrates in chest X-ray, ARDS or sepsis/septic shock. All patients received protocolized pharmacological and supportive treatment after admission, and VTE prophylaxis with low molecular weight heparin. Demographic data and medical history of arterial hypertension, diabetes mellitus, smoking habit, obesity [body mass index (BMI) ≥30 kg/m$^2$], renal failure [estimated glomerular filtration rate (eGFR) by CKD-EPI <60 ml/min/1.73m$^2$], cardiovascular diseases and pulmonary diseases were obtained. Cardiovascular diseases included arrhythmia, congestive heart failure, ischemic heart disease, valvulopathy and hypertensive cardiomyopathy. Pulmonary diseases included chronic obstructive pulmonary disease, asthma, obstructive sleep apnea and pulmonary tuberculosis. Patients were considered to have thrombocytopenia when platelet count was lower than 140 x10$^9$/l, prolonged PT when PT was higher than 14 seconds, and elevated ferritin when serum ferritin levels were higher than 400 ng/ml.

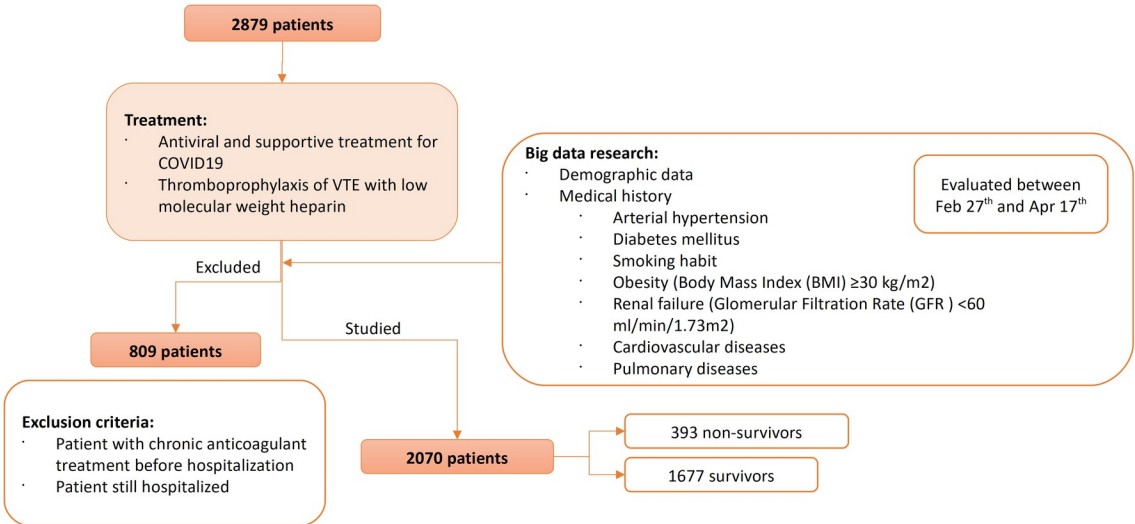

**Fig 1. Flow diagram of the sample selection and study design.**

Data were obtained from a big data research using extract transform load (ETL) tools and natural language processing (NLP) with our Huawei (Huawei Technologies Co., Ltd., Shenzhen, China) platform and the collaboration of Indizen-Scalian (Madrid, Spain). The clinical outcomes were monitored up to April 17, 2020. Only those patients that had been discharged from hospital or those who had died were finally recruited. Exclusion criteria: patients who remained hospitalized at the time of analysis and patients on chronic anticoagulant treatment before hospitalization. A flow diagram of the sample selection and study design is shown in Fig 1. The diagnosis of COVID-19 was made according to World Health Organization interim guidance [21] and confirmed by RNA detection of the 2019-nCoV in the clinical laboratory of Hospital Universitario Fundación Jiménez Díaz.

## Laboratory tests

D-dimer levels were determined on ACL Top 700 analyzer (Instrumentation Laboratory, Bedford, MA, USA) using a highly sensitive assay (IL D-dimer HS 500). Prothrombin time (PT), activated partial thromboplastin time (APTT), and fibrinogen were also determined on ACL Top 700 analyzer. Complete blood count was determined on Sysmex XN-1000 analyzer (Sysmex, Kobe, Japan). Serum ferritin levels were determined on Roche Cobas 6000 (Roche Diagnostics, Mannheim, Germany).

## Ethics statement

This observational study followed the ethical principles of the Helsinki Declaration and was previously approved by the Ethics Committee for Clinical Research of the Hospital Universitario Fundación Jiménez Díaz on April 14, 2020. Medical records of all the patients included were accessed from April 1 to May 15, 2020. All data were fully anonymized before we accessed them. Due to the retrospective nature of our study, the ethics committee waived the requirement for informed consent.

## Statistical analysis

All the laboratory results analyzed (serum ferritin, lymphocyte count, APTT, PT, fibrinogen levels, D-dimer levels, and platelet count) were the first determination of each parameter, which had been performed either in the emergency department or within 3 days from admission to ward. Age, gender and chronic comorbidities (arterial hypertension, diabetes mellitus, obesity, smoking habit, renal failure, cardiovascular disease and pulmonary disease) were also analyzed. Statistical comparisons of survivors and non-survivors were calculated using the chi-square test for categorical variables and Student's t test for continuous variables. The results were expressed as mean ± standard deviation if normal distributed, and as median (25–75 percentiles) if skewed, and numbers (percentage). Two-sided $p$ values less than 0.05 were considered statistically significant.

In order to simplify the score and increase its reproducibility and applicability in other hospitals and countries, the statistically significant quantitative variables were categorized. Age was splitted into 5 subgroups ($\leq$50, 51–60, 61–70, 71–80, and >80 years-old) since it is the most important prognostic factor [5]. We applied a previously published cut-off for D-dimer levels $\leq$1000 μg/l [two-fold increase of upper limit of normality (ULN)] or >2 ULN [3], whereas the other two variables were categorized into two subgroups according to their normality range: PT $\leq$14 or >14 seconds and platelet count $\leq$140 x$10^9$/l or >140 x$10^9$/l.

Statistically significant variables in the categorical analysis were included in a logistic regression model, performed in a randomly selected training cohort including around 60% of the total amount of patients. Missing data were estimated by multiple imputation with 50 different estimations performed [22]. Enter method was employed with Wald $P$ values. In order to achieve a better adjustment of the model, once significant variables were identified, a new model including only these variables was estimated. Logistic regression coefficients and $P$ values shown were obtained from the pooled analysis. Brier score analysis was calculated and odds predicted by the model were analyzed by using a ROC analysis. Prognostic features of the model in both cohorts were calculated by using a complete case analysis. A cut-off was selected based on its sensitivity and specificity. The logistic regression coefficients and the cut-off selected were validated in a different cohort composed of around 40% of total patients. Sensitivity, specificity, and predictive values in both cohorts were assessed and two-sided confidence intervals (CI) were calculated by the Wilson method. This was carried out using the Domenech Macro! DTfor SPSS (http://www.metodo.uab.cat/macros.htm). All statistical tests were performed in SPSS version 19.0 statistics package.

We adhered to the transparent reporting of a multivariable prediction model for individual prognosis or diagnosis (TRIPOD) statement for reporting [23].

## Results

A total of 2879 moderate to severe COVID-19 hospitalized patients were initially evaluated for inclusion in the development cohort. Of these, 809 were excluded: 515 remained hospitalized at the time of analysis and 294 were on chronic anticoagulant treatment before hospitalization. The final sample consisted of 2070 patients (884 females and 1186 males) with definite outcomes: 1677 (81.01%) patients had been discharged (survivors) and 393 (18.99%) patients had died (non-survivors).

The laboratory parameters and clinical characteristics of the patients at baseline are presented in Table 1 for all patients, survivors and non-survivors; data on some variables were missing for some patients. The mean age at disease onset was 65.68 years (range, 20–104). The proportion of male patients was higher in non-survivors (20.92% *vs*. 16.29%, $p$ = 0.008). The

**Table 1. Baseline characteristics of survivors and non-survivors, and univariate analysis.**

| Parameters | Total (n = 2070) | Survivors (n = 1677) | Non-survivors (n = 393) | P values | Odds ratio (95% CI) |
|---|---|---|---|---|---|
| Age at diagnosis (years) | 67 (54–79) | 63 (51–75) | 81 (72–87) | <0.001 | |
| Age categorized | | | | | |
| ≤50 | 403 (19.48) | 392 (97.27) | 11 (2.73) | <0.001 | |
| 51–60 | 379 (18.32) | 360 (95.00) | 19 (5.00) | <0.001 | 1.88 (0.88–4.01) |
| 61–70 | 413 (19.96) | 363 (89.89) | 50 (12.11) | <0.001 | 4.91 (2.52–9.57) |
| 71–80 | 437 (21.12) | 325 (74.37) | 112 (25.63) | <0.001 | 12.28 (6.49–23.21) |
| >80 | 437 (21.12) | 237 (54.23) | 200 (45.77) | <0.001 | 30.07 (16.05–56.35) |
| Gender | | | | | |
| Female | 884 (42.71) | 740 (83.71) | 144 (16.29) | 0.008 | |
| Male | 1186 (57.29) | 938 (79.08) | 248 (20.92) | 0.008 | 1.36 (1.08–1.70) |
| Comorbidities- n (%) | | | | | |
| No | 752 (36.32) | 686 (91.20) | 66 (8.80) | <0.001 | |
| Yes | 1318 (63.68) | 992 (75.30) | 326 (24.70) | <0.001 | 3.42 (2.58–4.53) |
| Arterial hypertension- n (%) | | | | | |
| No | 1140 (55.37) | 1006 (88.24) | 134 (11.76) | <0.001 | |
| Yes | 919 (44.63) | 664 (72.25) | 255 (27.75) | <0.001 | 2.88 (2.29–3.63) |
| Diabetes mellitus- n (%) | | | | | |
| No | 1650 (80.13) | 1380 (83.63) | 270 (16.37) | <0.001 | |
| Yes | 409 (19.87) | 290 (70.90) | 119 (29.10) | <0.001 | 2.09 (1.63–2.69) |
| Obesity- n (%) | | | | | |
| No | 707 (70.56) | 588 (83.17) | 119 (16.83) | 0.664 | |
| Yes | 295 (29.44) | 242 (82.03) | 53 (17.97) | 0.664 | 1.08 (0.76–1.55) |
| Smoking- n (%) | | | | | |
| No | 1971 (95.86) | 1599 (81.13) | 372 (18.87) | 0.795 | |
| Yes | 85 (4.14) | 68 (80.00) | 17 (20.00) | 0.795 | 1.07 (0.62–1.85) |
| Cardiovascular disease- n (%) | | | | | |
| No | 1732 (84.24) | 1446 (83.49) | 286 (16.51) | <0.001 | |
| Yes | 324 (15.76) | 222 (68.51) | 102 (31.49) | <0.001 | 2.32 (1.78–3.03) |
| Pulmonary disease- n (%) | | | | | |
| No | 1716 (83.42) | 1406 (81.93) | 310 (18.07) | 0.028 | |
| Yes | 341 (16.58) | 262 (76.83) | 79 (23.17) | 0.028 | 1.37 (1.03–1.81) |
| Chronic kidney disease- n (%) | | | | | |
| No | 1665 (95.91) | 1439 (86.42) | 226 (13.58) | <0.001 | |
| Yes | 71 (4.09) | 42 (59.15) | 29 (40.85) | <0.001 | 4.39 (2.68–7.20) |
| Lymphocyte count (x10$^9$/l) (NR 1.2–5) | 1.00 (0.70–1.30) | 1.00 (0.70–1.40) | 0.80 (0.52–1.10) | 0.539 | |
| Platelet count (x10$^9$/l) (NR 140–450) | 209.00 (161.00–274.00) | 213.00 (104.00–277.25) | 195.50 (145.25–256.75) | <0.001 | |
| Thrombocytopenia | | | | | |
| No | 1701 (84.62) | 1411 (83.00) | 290 (17.00) | <0.001 | |
| Yes | 309 (15.38) | 223 (72.16) | 86 (27.84) | <0.001 | 1.87 (1.42–2.48) |
| PT (seconds) (NR 10–14) | 12.80 (12.10–13.80) | 12.80 (12.10–13.70) | 13.20 (12.40–14.30) | 0.001 | |
| Prolonged PT | | | | | |
| No | 1517 (80.35) | 1264 (83.32) | 253 (16.68) | <0.001 | |
| Yes | 371 (19.65) | 273 (73.58) | 98 (26.42) | <0.001 | 1.79 (1.37–2.34) |
| APTT (seconds) (NR 26–36) | 30.40 (28.10–32.70) | 30.50 (28.20–32.70) | 29.80 (27.40–33.00) | 0.549 | |
| Fibrinogen (mg/dl) (NR 200–400) | 677.00 (568.00–801.00) | 679.00 (570.00–801.00) | 677.00 (564.00–805.00) | 0.220 | |
| D-dimer levels (μg/l) (NR 70–500) | 623.50 (336.00–1106.50) | 562.00 (315.04–995.00) | 1046.00 (568.35–1976.00) | 0.001 | |
| Elevated D-dimer (>2 ULN) | | | | | |

(*Continued*)

**Table 1.** (Continued)

| Parameters | Total (n = 2070) | Survivors (n = 1677) | Non-survivors (n = 393) | P values | Odds ratio (95% CI) |
|---|---|---|---|---|---|
| No | 1185 (70.62) | 1052 (88.77) | 133 (11.23) | <0.001 | |
| Yes | 493 (29.38) | 343 (69.57) | 150 (30.43) | <0.001 | 3.46 (2.66–4.50) |
| Ferritin levels (ng/ml) (NR 30–400) | 613.00 (314.00–1294.50) | 626.50 (354.50–1313.50) | 501.50 (241.00–1145.00) | 0.028 | |
| Ferritin elevated | | | | | |
| No | 373 (28.15) | 313 (83.91) | 60 (16.09) | 0.883 | |
| Yes | 952 (71.85) | 802 (84.24) | 150 (15.76) | 0.883 | 0.97 (0.70–1.35) |

APTT = activated partial Thromboplastin time; CI = confidence interval; dl = decilitre; l = litre; mg = miligrams; ml = mililitre; ng = nanograms; NR = normal range; PT = prothrombin time; ULN = upper limit of normal range. Missing data: age (1), lymphocyte count (60), platelet count (60), PT (182), APTT (176), fibrinogen (216), D-dimer levels (392), ferritin levels (1742).

mean length of hospital stay was 6.87 days (range, 0–41) in survivors and 6.51 days (range, 0–35) in non-survivors.

Compared with survivors, non-survivors showed higher D-dimer levels on admission, prolonged PT and lower platelet count (Table 1). No significant differences were found in smoking, obesity, lymphocyte count, APTT and fibrinogen levels. Additionally, there were no differences in the duration of hospitalization between survivors and non-survivors (6.87 ± 5.86 days *vs.* 6.51 ± 5.25 days, $p = 0.232$).

Significant differences in their in-hospital mortality were observed in categorized quantitative variables. In-hospital mortality was 2.73% in patients younger than 50 years old (used as the reference category) ($p < 0.001$), 5% in those between 51 and 60 years [Odds ratio (OR) 1.88; 95% CI, 0.88 to 4.01], 12.11% in those between 61 and 70 years (OR 4.91; 95% CI, 2.52 to 9.57), 25.63% in the 71–80 group (OR 12.28; 95% CI, 6.49 to 23.21), and 45.77% in those older than 80 years old (OR 30.07; 95% CI, 16.05 to 56.35). The proportion of non-survivors was significantly higher in COVID-19 patients with on-admission D-dimer levels >2 ULN ($p < 0.001$; OR 3.46; 95% CI, 2.66 to 4.50), prolonged PT ($p < 0.001$; OR 1.79; 95% CI, 1.37 to 2.34), and thrombocytopenia ($p < 0.001$; OR 1.87; 95% CI, 1.42 to 2.48). Additionally, OR of in-hospital mortality was higher in patients with arterial hypertension, cardiovascular diseases, pulmonary diseases, and renal failure. Although non-survivors had slightly lower serum ferritin levels, when categorized according to elevated ferritin (yes/no), no differences were found between both groups.

A total of ten parameters showed statistically significant differences between survivors and non-survivors. They were then examined in a multivariate logistic regression model including 1270 patients to identify independent prognostic factors of moderate/severe COVID-19 in-hospital mortality (Table 2). The following features were identified as independent predictors of poor outcome on multivariable analysis: age 61–70 years ($p < 0.001$; OR: 7.69; 95%CI: 2.93 to 20.14), age 71–80 years ($p < 0.001$; OR: 14.99; 95%CI: 5.88 to 38.22), age >80 years ($p < 0.001$; OR: 36.78; 95%CI: 14.42 to 93.85), male gender ($p < 0.001$; OR: 1.84; 95%CI: 1.31 to 2.58), D-dimer levels >2 ULN ($p = 0.003$; OR: 1.79; 95%CI: 1.22 to 1.62), and prolonged PT ($p < 0.001$; OR: 2.18; 95%CI: 1.49 to 3.18) (Table 2). Arterial hypertension, diabetes mellitus, cardiovascular disease, pulmonary disease, renal failure, and thrombocytopenia lost their significance and were not included in the final model. D-dimer levels >1000 μg/l (2 ULN), prolonged PT, male gender, and age showed an increase in the probabilities of death. The model showed no overdispersion. The formula of MRS-COVID-19 (Mortality Risk prognostic Score for hospitalized

**Table 2. Multivariate analysis.**

| Parameters | P values | Odds ratio (95% CI) |
|---|---|---|
| Age categorized (years) * | | |
| 51–60 | 0.115 | 2.35 (0.81–6.82) |
| 61–70 | <0.001 | 7.69 (2.93–20.14) |
| 71–80 | <0.001 | 14.99 (5.88–38.22) |
| >80 | <0.001 | 36.78 (14.42–93.85) |
| Gender (male) | <0.001 | 1.84 (1.31–2.58) |
| Arterial hypertension† | 0.705 | 0.93 (0.64–1.35) |
| Diabetes mellitus† | 0.251 | 1.24 (0.86–1.78) |
| Pulmonary disease† | 0.303 | 1.24 (0.83–1.85) |
| Cardiovascular disease† | 0.945 | 1.01 (0.68–1.52) |
| Chronic kidney disease† | 0.455 | 1.31 (0.65–2.64) |
| Thrombocytopenia† | 0.215 | 1.29 (0.86–1.93) |
| Prolonged PT | <0.001 | 2.18 (1.49–3.18) |
| D-dimer elevated (>2 ULN) | 0.003 | 1.79 (1.22–2.62) |
| Constant | <0.001 | 0.01 (0.00–0.03) |

* Age ≤50 years was used as the reference category.

† Variable not included in the final model.

COVID-19 patients) is:

$$\text{MRS-COVID-19} = \text{EXP}\{-4.585 + (0.610 \text{ if male gender}) + (0.581 \text{ if D-dimer} > 1000) + (0.778 \text{ if prolonged PT}) + [(0.855 \text{ if age } 51\text{-}60)$$

$$\text{OR } (2.04 \text{ if age } 61\text{-}70) \text{ OR } (2.708 \text{ if age } 71\text{-}80) \text{ OR } (3.605 \text{ if age} > 80)]\}.$$

A cut-off of 0.076 was arbitrarily selected, in order to maximize sensitivity and negative predictive value. In the first cohort (n = 1270; missing data = 270), an AUC of 0.81 was obtained, with a sensitivity of 95.83% (95% CI, 91.65 to 97.97), a specificity of 41.46% (95% CI, 38.16 to 44.85), negative predictive value (NPV) of 98.01% (95% CI, 95.95 to 99.03), and a positive predictive value (PPV) of 24.85% (95% CI, 21.67 to 28.31). Mortality rate in this cohort was 16.81%. In the validation cohort (n = 800; missing data = 185), an AUC of 0.80 was obtained, with a sensitivity of 92.52% (95% CI, 85.94 to 96.16), a specificity of 41.34% (95% CI, 37.14 to 45.67), NPV of 96.33% (95% CI, 92.93 to 98.13), and a PPV of 24.94% (95% CI, 20.93 to 29.42) (Table 3). Mortality rate in the validation cohort was 17.39%, comparable to first cohort´s. Brier score was 0.11 and 0.12 in the development and validation cohorts, respectively.

An interactive risk calculator for the application of individual combinations of the five parameters is provided at GooglePlay called MRS-COVID-19. This calculator allows for the classification of patients into low-risk or high-risk of in-hospital mortality and estimates OR values using young female without coagulopathy markers as the reference category.

## Discussion

The main finding of our study was the development and validation of a predictive model of in-hospital mortality based on age, gender, and on-admission coagulopathy markers of COVID-19. The actual COVID-19 pandemic has become a huge challenge for the health care systems of many countries due to the massive number of infected subjects. Emergency departments

**Table 3. Comparison of the distribution of parameters, AUC, sensitivity, specificity, PPV and NPV in the development and validation cohorts of MRS-COVID-19 score.**

| | Development cohort n = 1270 | Validation cohort n = 800 |
|---|---|---|
| Patients with missing data | 270 | 185 |
| AUC | 0.81 | 0.80 |
| Sensitivity -% (95% CI) | 95.83 (91.65–97.97) | 92.52 (85.94–96.16) |
| Specificity -% (95% CI) | 41.46 (38.16–44.85) | 41.34 (37.14–45.67) |
| PPV -% (95% CI) | 24.85 (21.67–28.31) | 24.94 (20.93–29.42) |
| NPV -% (95% CI) | 98.01 (95.95–99.03) | 96.33 (92.93–98.13) |
| Mortality (%) | 16.81 | 17.39 |
| Age categorized- n (%) | | |
| ≤50 | 244 (19.23) | 159 (19.88) |
| 51–60 | 245 (19.32) | 134 (16.75) |
| 61–70 | 238 (18.75) | 175 (21.87) |
| 71–80 | 273 (21.51) | 164 (20.50) |
| >80 | 269 (21.19) | 168 (21.00) |
| Gender- n (%) | | |
| Male | 729 (57.40) | 457 (57.12) |
| Female | 541 (42.60) | 343 (42.88) |
| Arterial hypertension- n (%) | | |
| No | 694 (54.86) | 446 (56.17) |
| Yes | 571 (45.14) | 348 (43.83) |
| Diabetes mellitus- n (%) | | |
| No | 999 (78.97) | 651 (81.98) |
| Yes | 266 (21.03) | 143 (18.02) |
| Pulmonary disease- n (%) | | |
| No | 1049 (83.05) | 667 (84.01) |
| Yes | 214 (16.95) | 127 (15.99) |
| Cardiovascular disease- n (%) | | |
| No | 1061 (84.07) | 671 (84.51) |
| Yes | 201 (15.93) | 123 (15.49) |
| Renal Failure- n (%) | | |
| No | 1011 (96.19) | 654 (95.47) |
| Yes | 40 (3.81) | 31 (3.53) |
| Thrombocytopenia- n (%) | | |
| No | 1048 (84.17) | 565 (79.58) |
| Yes | 197 (15.83) | 137 (20.42) |
| Prolonged PT- n (%) | | |
| No | 952 (80.81) | 565 (79.58) |
| Yes | 226 (19.19) | 145 (20.42) |
| D-dimer elevated (>2 ULN)- n (%) | | |
| No | 728 (70.47) | 457 (70.85) |
| Yes | 305 (29.53) | 188 (29.15) |

AUC = area under the curve; NPV = negative predictive value; PPV = positive predictive value; ULN = upper limit of normality.

have been overwhelmed due to insufficient medical personnel and resources and patient over-crowding [24]. The access to invasive ventilation and/or intensive care units has been limited or prioritized to patients developing severe hypoxemic respiratory failure. In order to address these shortages and their consequences, it is essential that health care systems develop efficient strategies and plans to effectively deal with them. A risk model or score capable of predicting on admission which COVID-19 patients will most probably survive would be a strategy of great interest, in order to avoid the collapse of acute care hospitals as far as possible. Thus, predictive models with high sensibility and, therefore, high negative predictive value would be desirable, since low-risk patients could either be discharged or derived to other support institutions that lack intensive care units.

Our study demonstrates that in-hospital mortality among moderate or severe hospitalized COVID-19 patients is predicted by the combination of age, gender, and coagulopathy markers (D-dimer and PT). The regression coefficients and cut-off selected were then validated in an independent data sample. Because our aim was to create a screening tool, we intentionally used a cut-off with high sensitivity and NPV, but low specificity and PPV. Therefore, the proportion of patients misclassified as high-risk will be elevated. However, patients classified as low-risk on admission could get either discharged early or derived to other centers without intensive care units with the certainty that their likelihood of dying is not as high as those classified as high risk, based on our arbitrarily selected and afterwards validated cut-off. Further external validation of our findings should be performed. Although COVID-19 mortality rates may be lower in future outbreaks due to improvements in its management and better access to medical infrastructures, the predictive capacity of our model should not be worse.

The model could be easily implemented in any laboratory information system (LIS), so that clinicians may automatically have the prognostic information. Additionally, in clinical trials that include adult COVID-19 patients of all ages, our model could be useful to ensure the comparability of included comparison groups.

Based on the logistic regression model coefficients, age was confirmed to be the strongest predictor of mortality in our cohort. Most of COVID-19 patients aged less than 50 years old (97.27%) or between 51 and 60 years old (95%) will be discharged within a few days regardless of their laboratory parameters on admission. On the other hand, near half of the patients over 80 years old died (45.77%), probably owing to a less rigorous immune response, thus suggesting that our predictive model seems to be less helpful in extreme age ranges. However, the addition of coagulopathy markers to age and gender may help clinicians refine the prognosis of hospitalized COVID-19 patients, especially those aged between 50 and 70 years.

Moderate or severe COVID cases are more likely to occur in older men with comorbidities [1]. A recent meta-analysis with aggregated data, including a total number of 3027 COVID-19 patients, confirmed that male, aged over 65 years, smoking and comorbidities such as hypertension, diabetes, cardiovascular disease, and respiratory diseases were risk factors for severe disease and mortality [2]. More than 60% of our 2070 cases were over 60 years old, and the likelihood of dying was higher in men compared to women. Non-survivors from our cohort were older, had more chronic pathologies (with the exception of obesity and smoking habit), and a showed a higher proportion of males. Our findings are in agreement with previous reports, since the outcome was significantly worse in male patients and those with chronic pathologies. However, the presence of all of these comorbidities was excluded from our final model.

Although the pathophysiology underlying severe COVID-19 remains poorly understood, a lung-centric coagulopathy is believed to play an important role [25]. COVID-19 associated coagulopathy correlates with illness severity and mortality, and may include increased D-dimer levels, mild PT prolongation and mild thrombocytopenia [10, 13, 26]. Thrombotic

complications have emerged as an important issue in COVID-19 patients as a result of the inflammatory response to SARS-CoV-2. COVID-19 prothrombotic status seems to be multi-factorial. The illness severity and hypoxia, hemostatic abnormalities, the severe inflammatory response, plus any other underlying thrombotic risk factors can lead to a thrombotic event [27].

Compared to survivors, the COVID-19 non-survivors from our cohort presented significantly higher D-dimer levels, prolonged PT and lower platelet counts. These results are in agreement with previously published data [10–14]. Similar to our approach, Zhang et al retrospectively analyzed 343 COVID-19 hospitalized patients and reported that a four-fold increase of on-admission D-dimer levels could effectively predict their in-hospital mortality [20]. To our knowledge, there are two studies reporting predictive models of mortality in adult hospitalized COVID-19 patients based on baseline clinical and laboratory data [28, 29]. Their risk of bias is high, either because the sample size is small or because they are not validated.

Wang and colleagues developed (n = 296) and validated (n = 44) two models, both based on age: one clinical (including age, hypertension and coronary heart disease sensitivity), and one based on laboratory parameters [age, C-reactive protein, $SpO_2$, neutrophil and lymphocyte count, D-dimer, aspartate aminotransferase (AST) and GFR] which had a significantly stronger discriminatory power than the clinical model [28]. The model from Chen and colleagues was developed from a bigger retrospective cohort (n = 1590), and included age, coronary heart disease, cerebrovascular disease, dyspnea, procalcitonin level >0.5 ng/mL, and AST) [29]. However, it has not been validated.

Although most of predictive models have been reported to be at high risk of bias [30], we adhered to the TRIPOD reporting guideline [23] to perform our model, and the Brier test results ensure its good calibration.

The strengths of our model include the study population size, the multicenter nature of data and the inclusion of a validation cohort. However, the model has some limitations. First of all, it is a retrospective analysis. Second, no data from possible hospital readmission of survivors were available, and it is possible that some initially recovered patients may have worsened a few days later. Finally, although we obtained dichotomized variables in order to simplify the model and increase its applicability, the use of continuous variables has the potential to provide more refined information.

In conclusion, we developed and validated a predictive model for in-hospital mortality of moderate or severe COVID-19 patients, which included D-dimer levels >2 ULN, prolonged PT, male gender and age as positively correlated factors with fatal outcome. Our findings, obtained and validated from a large series of hospitalized COVID-19 patients, support the use of this prognostic tool on admission to identify a low-risk group that may benefit from early discharge or derivation to support institutions, in order to prevent acute care hospitals getting overwhelmed. Prospective studies are needed to confirm our findings.

## Supporting information

**S1 Data.**
(XLSX)

## Acknowledgments

We acknowledge all health-care workers involved in the diagnosis and treatment of COVID-19 patients in Hospital General de Villalba (Collado Villalba, Madrid), Hospital Infanta Elena

(Valdemoro, Madrid), Hospital Universitario Rey Juan Carlos (Móstoles, Madrid) and Hospital Universitario Fundación Jiménez Díaz in Madrid.

## Author Contributions

**Conceptualization:** Diego Velasco-Rodríguez, Rosa Vidal Laso, Pilar Llamas-Sillero.

**Data curation:** Juan-Manuel Alonso-Dominguez, Antonio Herrero.

**Formal analysis:** Juan-Manuel Alonso-Dominguez, Antonio Herrero.

**Methodology:** Diego Velasco-Rodríguez, Juan-Manuel Alonso-Dominguez, Rosa Vidal Laso.

**Project administration:** Pilar Llamas-Sillero.

**Software:** Daniel Lainez-González.

**Supervision:** Pilar Llamas-Sillero.

**Validation:** Diego Velasco-Rodríguez, Juan-Manuel Alonso-Dominguez, Rosa Vidal Laso, Daniel Lainez-González, Aránzazu García-Raso, Sara Martín-Herrero, Inés Martínez Alfonzo, Juana Serrano-López, Elena Jiménez-Barral, Sara Nistal, Manuel Pérez Márquez, Elham Askari, Jorge Castillo Álvarez, Antonio Núñez, Ángel Jiménez Rodríguez, Sarah Heili-Frades, César Pérez-Calvo, Miguel Górgolas, Raquel Barba, Pilar Llamas-Sillero.

**Visualization:** Rosa Vidal Laso, Pilar Llamas-Sillero.

**Writing – original draft:** Diego Velasco-Rodríguez, Juan-Manuel Alonso-Dominguez, Rosa Vidal Laso.

**Writing – review & editing:** Diego Velasco-Rodríguez, Juan-Manuel Alonso-Dominguez, Rosa Vidal Laso, Pilar Llamas-Sillero.

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
