## [Decision Letter · Decision Letter 0]

21 Dec 2020

PONE-D-20-35733

DEVELOPMENT AND VALIDATION OF A PREDICTIVE MODEL OF IN-HOSPITAL MORTALITY IN COVID-19 PATIENTS

PLOS ONE

Dear Dr. Alonso-Dominguez,

Thank you for submitting your manuscript to PLOS ONE. After careful consideration, we feel that it has merit but does not fully meet PLOS ONE’s publication criteria as it currently stands. Therefore, we invite you to submit a revised version of the manuscript that addresses the points raised during the review process.

We look forward to receiving your revised manuscript.

Kind regards,

Aleksandar R. Zivkovic

Academic Editor

PLOS ONE

Journal Requirements:

2. In the ethics statement in the manuscript and in the online submission form, please provide additional information about the patient records/samples used in your retrospective study, including the date range (month and year) during which patients' medical records/samples were accessed.

3. In your ethics statement in the manuscript and in the online submission form, please ensure that you have discussed whether all data/samples were fully anonymized before you accessed them and/or whether the IRB or ethics committee waived the requirement for informed consent. If patients provided informed written consent to have data/samples from their medical records used in research, please include this information.

4.In your Data Availability statement, you have not specified where the minimal data set underlying the results described in your manuscript can be found. PLOS defines a study's minimal data set as the underlying data used to reach the conclusions drawn in the manuscript and any additional data required to replicate the reported study findings in their entirety. All PLOS journals require that the minimal data set be made fully available. For more information about our data policy, please see http://journals.plos.org/plosone/s/data-availability.

5.Thank you for stating the following in the Financial Disclosure section:

We note that one or more of the authors are employed by a commercial company: Quironsalud

Reviewers' comments:

Reviewer's Responses to Questions

**Comments to the Author**

1. Is the manuscript technically sound, and do the data support the conclusions?

Reviewer #1: Yes

Reviewer #2: Yes

Reviewer #3: Partly

Reviewer #4: Partly

2. Has the statistical analysis been performed appropriately and rigorously? 

Reviewer #1: Yes

Reviewer #2: Yes

Reviewer #3: No

Reviewer #4: Yes

3. Have the authors made all data underlying the findings in their manuscript fully available?

Reviewer #1: Yes

Reviewer #2: Yes

Reviewer #3: Yes

Reviewer #4: No

4. Is the manuscript presented in an intelligible fashion and written in standard English?

Reviewer #1: Yes

Reviewer #2: Yes

Reviewer #3: Yes

Reviewer #4: No

5. Review Comments to the Author

Reviewer #1: In this retrospective observational multicenter study, the authors developed and validated a predictive model for in-hospital mortality of moderate and severe COVID-19 patients, using demographic data, medical history and laboratory parameters. Overall, the study was well executed and the authors achieved its clearly defined objectives. The manuscript was concise, easy to understand and a good read. It contributes to the existing literature on prognostication of outcomes in COVID-19 patients. Following are listed some comments and points to be addressed:

Major issues:

---------------

1) The authors states (p. 19, l. 319) that a predictive model with a high negative predictive value is desirable, since this allows to identify low-risk patients which may be discharged at an earlier stage, and thereby relieve the health care system. However, since the data from the present study is based on patients (both survivors and non-survivors) receiving full treatment, I could fear that mortality will increase in the low-risk patients if they are discharged early solely based on their risk score. The authors should emphasize the need for prospective studies to back up this statement in the conclusions.

2) Was there a sample size analysis or was this a sample of convenience?

Minor issues:

---------------

1) The authors should either combine the first two sentences or rewrite the first sentence in the abstract (p. 2, l. 85-88).

2) In the statistical analysis section (p. 6, l. 190-192), the authors state: “The results were expressed as the mean ± standard deviation and range or number (percentage), wherever appropriate. P values less than 0.05 were considered statistically significant.” It would be more informative and logic to present data as mean ± standard deviation if normal distributed, and as median (25-75 percentiles) if skewed, and numbers (percentage). Even though the authors states that range is listed, I think the authors have instead presented minimum and maximum values, which is of less importance. Also, it should be stated if the P values used are one- or two-sided.

3) Table 1 is very long and difficult to overlook. I would suggest removing ranges, and instead list mean ± SD if normal distributed, median (25-75 percentiles) if skewed, and number (percentage). In the column listing OR (95% CI), please type reference group in each of the univariate analyses involving categorical data, starting with the reference group. It is not clear which variable is associated with a higher OR (e.g. going from male to female, or female to male). Missing data of the individual parameters should be listed as a foot note. Also, the category “hospital stay” is somehow relevant and informative but also misleading, since non-survivors are discharged to the morgue. The authors should remove this parameter in Table 1, and carefully mention the results in the Results section. Finally, “sex” is used in the table, but “gender” is used in the text. Please be consistent.

4) Results are listed in details both in the tables and in the Results section. Data should generally only be listed either in the tables or in the text. Please refine manuscript.

5) Serum ferritin have in several studies been found valuable to determine poor prognosis in COVID-19 patients. The authors state (p. 14, l. 259); “No significant differences were found in categorized ferritin”. Looking at Table 1, non-survivors had significantly higher serum ferritin levels, but when categorized according to elevated ferritin (yes/no), this is no longer significantly different between groups. This detail should be added to the text. Also, please define the term “elevated ferritin” in the Methods section.

6) The authors used area under the ROC curve to evaluate both the development model and the validation model. Even though this is a generally accepted method in medical literature, the AUC only describe the discriminative ability of the model, i.e. correctly allocate patients as survivor or non-survivors according the model. However, adding a statistical analysis to test the predictive ability, e.g. Brier scores, could provide additional strength to the study.

Reviewer #2: Dear Authors;

I have read the manuscript titled “DEVELOPMENT AND VALIDATION OF A PREDICTIVE MODEL OF IN-HOSPITAL MORTALITY IN COVID-19 PATIENTS”. This study retrospectively evaluated 2879 hospitalized COVID-19 patients from four hospitals. Although it is a nice designed study there are several limitations.

1. There are more than 50 studies investigating the in hospital or 7-14 days mortality in patients with COVID-19 infection. Therefore (unfortunately) this manuscript doesn’t add something new to the literature.

2. This new predictive model has a sensitivity of 95.83%, specificity of 41.46%, negative predictive value of 98.01%, and positive predictive value of 24.85%. The authors explain these results however for a predictive model specificity is quite low.

3. doi: https://doi.org/10.1136/bmj.m1328 is a nice review and summary of literature why these predictive models have bias.

4. One selection bias is the inclusion criteria which CURB-65 score is used. Literature showed that CALL or other scores may be used instead of CURB-65 for covid pneumonia.

5. The authors found no difference in lymphocyte count between deceased and survivors. An interesting finding because in most studies lymphocyte count is a predictive factor for mortality.

Thank you.

Reviewer #3: Thank you for the opportunity to review this manuscript, which focusses on an important topic and addresses the important issue of risk stratification for patients with suspected and confirmed Covid-19 infection. With regards to the model presented, I am encouraged to see a relatively large sample size and contemporary data from patients across several centres.

However, I have serious concerns about a number of methodological flaws with the development and internal validation of the model that make it impossible for me to recommend publication at this time:-

1. Inclusion of predictors. Predictors were initially selected to be included in the logistic regression model if a significant association was found between the predictor and the outcome on univariable analysis. This is recognised as an unsuitable approach and may lead to important predictors being excluded. [1] All predictors deemed to be of clinical significance should be included in the logistic regression analysis, regardless of the univariable statistical analysis.

2. Sample size. Although 2070 patients were included in the study, no formal statistical sample size analysis has been undertaken. [2] This is an important step to ensure that the results of the model development can be considered statistically robust.

3. Dichotomisation/categorisation of continuous variables. Transformation of continuous variables into non-continuous variables can reduce the power by the same amount as discarding approximately 1/3 of the data and should be avoided unless absolutely necessary. For variables with recognised cut-offs (i.e. haemoglobin values for anaemia) it can be an acceptable practice, but in this model, you have dichotomised four variables including age. I consider this a serious methodological flaw. [3]

4. Measures of model performance. Whilst you have provided information regarding the discrimination of the model, there are no reported measures of calibration. Whilst discrimination is important in terms of assessing how well the model discriminates between those with and without the outcome calibration is a measure of how closely model predicted outcomes match observed outcomes. Some measure of calibration (observed to expected ratio, calibration plot, calibration slope, calibration-in-the-large) must be included when presenting a model. [4]

5. Internal validation. Whilst an attempt to perform internal validation has been undertaken, the methodological approach is not appropriate. Using a split sample approach (splitting the population into a development and a validation set) is no longer recognised as an acceptable method of internal validation due to a number of drawbacks. Internal validation should be performed using cross-validation or bootstrapping

I think that the multicentre dataset and important nature of the question being asked is worthy of further investigation. However, I think that the entire methodological approach to model development needs to be re-assessed and a new analysis performed. A new paper should be submitted in the future if you are able to do this.

References

[1] Royston P, Moons KGM, Altman DG, Vergouwe Y. Prognosis and prognostic research: developing a prognostic model. BMJ 2009;338: b604.

[2] BMJ 2020;368:m441 doi: 10.1136/bmj.m441

[3] Royston P, Altman DG, Sauerbrei W. Dichotomizing continuous predictors in multiple regression: a bad idea. Stat Med 2006;25:127–41.

[4] Grant SW, Collins GS, Nashef SAM. Statistical Primer: developing and validating a risk prediction model. Eur J Cardiothorac Surg 2018;54:203–8

[5] Altman DG, Vergouwe Y, Royston P, Moons KGM. Prognosis and prog nostic research: validating a prognostic model. BMJ 2009;338:b605.

Reviewer #4: It is an important topic and interesstung as well. The authors provide a possible predictors of COVID-19 in-Hospital Mortality.

However the current work suffer many Major issues.

Good idea of your work. Relevant study. Introduction of your paper states not clearly the problem but underlines your study purpose.

Material and methods leave behind unanswered questions.

Results are not clearly structured and the table is way too long and confusing

In the discussion you go more into the background of your study (which could be partly mentioned in your introduction). You advice a new prediction tool for in hospital mortality. However

you did a validation in the same paper without background information of the idependent patient sample. ???? that is very much questionable for me….

Major issues:

1. what is the Definition of moderat to Severe COVID-19 Patients?

2. Lines 221 A total of 2879 moderate to severe COVID-19 hospitalized patients were initially

lines 222 evaluated for inclusion in the development cohort. Of these, 809 were excluded: 515

line 223 remained hospitalized at the time of analysis and 294 were on chronic anticoagulant

line 224 treatment before hospitalization. The final sample consisted of 2070 patients (884

line 225 females and 1186 males) with definite outcomes: 1677 (81.01%) patients had been

line 226 discharged (survivors) and 393 (18.99%) patients had died (non-survivors).

Exclusion of a large group of patients. Approximately 28,1 % of patient were excluded.

Follow up time is not clear. (or time to mortality.)?

3. it is not clear how many patients were mechanical ventilated?

4.lines 262- 264 " They were then examined in a multivariate logistic regression model including 1270 patients to identify independent prognostic factors of moderate/severe COVID-19 in-hospital mortality (Table 2)."

Not clear. Sub-analysis of 2070 patients? Why are the 800 patients excluded from this Analysis.

Is that the independent sample of patients about which we read in the abstract.? Not clear.

IF Yes: no characteristics of this patient population present.

5. Line 283 A cut-off of 0.076 was arbitrarily selected.

Please elaborate more why this cut off value and not another ?

6. If a major Revision is possible, the analysis must be checked by a statistician

Minor

1. The language need to be re-edited, there are many typos as well as weak sentences: lines 85-86 To evaluate the risk of demographic data, medical history, and on-admission laboratory parameters in hospitalized COVID-19 patients to predict mortality

(Not well written)

Lines 133-135 Clinicians need better predictors of which patients are prone to deteriorate rapidly or who may go on to die

(“predict Mortality” better) not go die!!!!

6. PLOS authors have the option to publish the peer review history of their article (what does this mean?). If published, this will include your full peer review and any attached files.

Reviewer #1: **Yes: **Sebastian Roed Rasmussen

Reviewer #2: No

Reviewer #3: No

Reviewer #4: No

---

## [Author Response · Author response to Decision Letter 0]

8 Feb 2021

Dear Editor and reviewers,

Thank you very much for your comments. We truly appreciate them. 

Here are our responses (in red).

Journal Requirements:

1. Please ensure that your manuscript meets PLOS ONE's style requirements, including those for file naming. We have modified the manuscript in order to meet PLOS ONE's style requirements.

2. In the ethics statement in the manuscript and in the online submission form, please provide additional information about the patient records/samples used in your retrospective study, including the date range (month and year) during which patients' medical records/samples were accessed. 

3. In your ethics statement in the manuscript and in the online submission form, please ensure that you have discussed whether all data/samples were fully anonymized before you accessed them and/or whether the IRB or ethics committee waived the requirement for informed consent. If patients provided informed written consent to have data/samples from their medical records used in research, please include this information. 

Data will be available from the corresponding author upon reasonable request.

5. Thank you for stating the following in the Financial Disclosure section:

We have added this sentence in the manuscript.

We note that one or more of the authors are employed by a commercial company: Quironsalud

We have included the mentioned statement.

2. Please also provide an updated Competing Interests Statement declaring this commercial affiliation along with any other relevant declarations relating to employment, consultancy, patents, products in development, or marketed products, etc. An updated Competing Interests Statement was added.

The following statement was included: "This does not alter our adherence to PLOS ONE policies on sharing data and materials.”

Both an updated Funding Statement and Competing Interests Statement were included in the cover letter.

To reviewers:

Reviewer #1: In this retrospective observational multicenter study, the authors developed and validated a predictive model for in-hospital mortality of moderate and severe COVID-19 patients, using demographic data, medical history and laboratory parameters. Overall, the study was well executed and the authors achieved its clearly defined objectives. The manuscript was concise, easy to understand and a good read. It contributes to the existing literature on prognostication of outcomes in COVID-19 patients. Thank you for your comments. Here are our answers.

Following are listed some comments and points to be addressed:

Major issues:

---------------

1) The authors states (p. 19, l. 319) that a predictive model with a high negative predictive value is desirable, since this allows to identify low-risk patients which may be discharged at an earlier stage, and thereby relieve the health care system. However, since the data from the present study is based on patients (both survivors and non-survivors) receiving full treatment, I could fear that mortality will increase in the low-risk patients if they are discharged early solely based on their risk score. The authors should emphasize the need for prospective studies to back up this statement in the conclusions. We do agree with your comment. To date, although some drugs like dexametasone have shown benefit, none of the applied treatments have dramatically changed the outcomes in COVID-19 patients. Nevertheless, supportive treatment received may have decreased mortality rate in hospitalized patients. We have emphasized the need for prospective studies to back up this statement in the conclusions.

2) Was there a sample size analysis or was this a sample of convenience? We used all the available data we had at the time of the analysis.

Minor issues:

---------------

1) The authors should either combine the first two sentences or rewrite the first sentence in the abstract (p. 2, l. 85-88). Amended.

2) In the statistical analysis section (p. 6, l. 190-192), the authors state: “The results were expressed as the mean ± standard deviation and range or number (percentage), wherever appropriate. P values less than 0.05 were considered statistically significant.” It would be more informative and logic to present data as mean ± standard deviation if normal distributed, and as median (25-75 percentiles) if skewed, and numbers (percentage). Even though the authors states that range is listed, I think the authors have instead presented minimum and maximum values, which is of less importance. Also, it should be stated if the P values used are one- or two-sided. We have now presented data as median (25-75 percentiles) since all of quantitative variables had skewed distribution, and numbers (percentage). P values are two-sided. We have modified the text. You are absolutely right about the definition of range. We have amended this issue in the manuscript.

3) Table 1 is very long and difficult to overlook. I would suggest removing ranges, and instead list mean ± SD if normal distributed, median (25-75 percentiles) if skewed, and number (percentage). In the column listing OR (95% CI), please type reference group in each of the univariate analyses involving categorical data, starting with the reference group. It is not clear which variable is associated with a higher OR (e.g. going from male to female, or female to male). Missing data of the individual parameters should be listed as a foot note. Also, the category “hospital stay” is somehow relevant and informative but also misleading, since non-survivors are discharged to the morgue. The authors should remove this parameter in Table 1, and carefully mention the results in the Results section. Finally, “sex” is used in the table, but “gender” is used in the text. Please be consistent. We have modified Table 1 following your suggestions. OR values have been deleted in the reference categories, and ranges have been removed. We have listed missing data as a footnote.

4) Results are listed in details both in the tables and in the Results section. Data should generally only be listed either in the tables or in the text. Please refine manuscript. We have refined the manuscript by removing numeric values of the parameters in the Results section.

5) Serum ferritin has in several studies been found valuable to determine poor prognosis in COVID-19 patients. The authors state (p. 14, l. 259); “No significant differences were found in categorized ferritin”. Looking at Table 1, non-survivors had significantly higher serum ferritin levels, but when categorized according to elevated ferritin (yes/no), this is no longer significantly different between groups. This detail should be added to the text. Also, please define the term “elevated ferritin” in the Methods section. We have added this detail to the text and defined the term “elevated ferritin” in the Methods section.

6) The authors used area under the ROC curve to evaluate both the development model and the validation model. Even though this is a generally accepted method in medical literature, the AUC only describe the discriminative ability of the model, i.e. correctly allocate patients as survivor or non-survivors according the model. However, adding a statistical analysis to test the predictive ability, e.g. Brier scores, could provide additional strength to the study. Following your recommendation, Brier score was performed to test the predictive ability of our model, showing a good calibration.

Reviewer #2: Dear Authors;

I have read the manuscript titled “DEVELOPMENT AND VALIDATION OF A PREDICTIVE MODEL OF IN-HOSPITAL MORTALITY IN COVID-19 PATIENTS”. This study retrospectively evaluated 2879 hospitalized COVID-19 patients from four hospitals. Although it is a nice designed study there are several limitations.

1. There are more than 50 studies investigating the in hospital or 7-14 days mortality in patients with COVID-19 infection. Therefore (unfortunately) this manuscript doesn’t add something new to the literature. Although there are numerous studies that have evaluated COVID-19 mortality, very few of them carried out an external validation, and calibration was rarely assessed. Our score is well calibrated, validated and very easy to apply. In the actual setting, cases are rapidly increasing again, and the feared “third wave” is becoming real. Tools like ours can be of great utility in order to prevent the collapse of emergency departments and acute care hospitals.

2. This new predictive model has a sensitivity of 95.83%, specificity of 41.46%, negative predictive value of 98.01%, and positive predictive value of 24.85%. The authors explain these results however for a predictive model specificity is quite low. We have chosen the threshold of the score in order to maximize the negative predictive value and sensitivity. Of course, our score is not perfect but we believe its applicable and useful.

3. doi: https://doi.org/10.1136/bmj.m1328 is a nice review and summary of literature why these predictive models have bias. We have included this reference and discussed it.

4. One selection bias is the inclusion criteria which CURB-65 score is used. Literature showed that CALL or other scores may be used instead of CURB-65 for covid pneumonia. CURB-65 was one of the hospitalization criteria for COVID-19 patients in our institution protocol. Although CALL score has been developed specifically for COVID-19 pneumonia, it was not available when we performed our study.

5. The authors found no difference in lymphocyte count between deceased and survivors. An interesting finding because in most studies lymphocyte count is a predictive factor for mortality. It may seem surprising since lymphopenia on admission has been associated with poor outcome in patients with COVID-19 in many studies. However, we found no differences between survivors and deceased.

Thank you.

Reviewer #3: Thank you for the opportunity to review this manuscript, which focusses on an important topic and addresses the important issue of risk stratification for patients with suspected and confirmed Covid-19 infection. With regards to the model presented, I am encouraged to see a relatively large sample size and contemporary data from patients across several centres.

However, I have serious concerns about a number of methodological flaws with the development and internal validation of the model that make it impossible for me to recommend publication at this time:-

Thank you very much for your comments. 

1. Inclusion of predictors. Predictors were initially selected to be included in the logistic regression model if a significant association was found between the predictor and the outcome on univariable analysis. This is recognised as an unsuitable approach and may lead to important predictors being excluded. [1] All predictors deemed to be of clinical significance should be included in the logistic regression analysis, regardless of the univariable statistical analysis. You are completely right. However, we have to keep in mind that COVID-19 is a novel disease and there are no completely established predictive factors that we should have included in the multivariable analysis.

2. Sample size. Although 2070 patients were included in the study, no formal statistical sample size analysis has been undertaken. [2] This is an important step to ensure that the results of the model development can be considered statistically robust. You are completely right and sample size calculation is the optimal procedure. Nevertheless, our sample size seems big enough taking into account the “10 events per predictor variable” rule. We preferred to include all the data we had available.

3. Dichotomisation/categorisation of continuous variables. Transformation of continuous variables into non-continuous variables can reduce the power by the same amount as discarding approximately 1/3 of the data and should be avoided unless absolutely necessary. For variables with recognised cut-offs (i.e. haemoglobin values for anaemia) it can be an acceptable practice, but in this model, you have dichotomised four variables including age. I consider this a serious methodological flaw. [3]

We have performed the dichotomization of laboratory variables in order to maximixe reproducibility of the score. Laboratory measurements as d-dimer, protrombine time, platelet count and ferritin might not yield the same results among different laboratories. By dichotomizing in normal or elevated valued we sought to increase the reproducibility, which is, in our opinion, especially interesting in the actual pandemic setting. 

We categorized age into several groups, as published in most studies including COVID-19 patients. Within these groups, the risk of mortality seems to be stable. We are aware that we can reduce the power of the score but by doing it in this way we can increase reproducibility of the score and easiness of use. We believe that both reproducibility and easiness are essential for a score to help to effectively deal with the shortage of resources during the actual COVID-19 pandemic, since the feared “third wave” is getting worse in Spain and many other countries.

4. Measures of model performance. Whilst you have provided information regarding the discrimination of the model, there are no reported measures of calibration. Whilst discrimination is important in terms of assessing how well the model discriminates between those with and without the outcome calibration is a measure of how closely model predicted outcomes match observed outcomes. Some measure of calibration (observed to expected ratio, calibration plot, calibration slope, calibration-in-the-large) must be included when presenting a model. [4] We agree with your comment. A predictive model needs to be calibrated. Brier score summarizes model calibration and discrimination. This test provides a measure of the agreement between the observed binary outcome and the predicted probability of that outcome. It was performed to evaluate how closely our model predicted outcomes match observed outcomes.

5. Internal validation. Whilst an attempt to perform internal validation has been undertaken, the methodological approach is not appropriate. Using a split sample approach (splitting the population into a development and a validation set) is no longer recognised as an acceptable method of internal validation due to a number of drawbacks. Internal validation should be performed using cross-validation or bootstrapping

Thank you for your comment. We guess you refer to recommendations included in reference 5. As it is stated in that manuscript “If the available data are limited, the model can be developed on the whole dataset and techniques of data re-use, such as cross validation and bootstrapping, applied to assess performance”. We consider our cohort of over 2000 patients is not limited. Therefore, applying bootstrapping or cross-validation techniques is not mandatory, and splitting the sample is a feasible approach.

I think that the multicentre dataset and important nature of the question being asked is worthy of further investigation. However, I think that the entire methodological approach to model development needs to be re-assessed and a new analysis performed. A new paper should be submitted in the future if you are able to do this.

References

[1] Royston P, Moons KGM, Altman DG, Vergouwe Y. Prognosis and prognostic research: developing a prognostic model. BMJ 2009;338: b604.

[2] BMJ 2020;368:m441 doi: 10.1136/bmj.m441

[3] Royston P, Altman DG, Sauerbrei W. Dichotomizing continuous predictors in multiple regression: a bad idea. Stat Med 2006;25:127–41.

[4] Grant SW, Collins GS, Nashef SAM. Statistical Primer: developing and validating a risk prediction model. Eur J Cardiothorac Surg 2018;54:203–8

[5] Altman DG, Vergouwe Y, Royston P, Moons KGM. Prognosis and prog nostic research: validating a prognostic model. BMJ 2009;338:b605.

Reviewer #4: It is an important topic and interesting as well. The authors provide possible predictors of COVID-19 in-Hospital Mortality.

However the current work suffers many Major issues.

Good idea of your work. Relevant study. Introduction of your paper states not clearly the problem but underlines your study purpose. Thank you for your suggestions. We have added the following sentence in the Introduction in order to clearly state the problem: “The COVID-19 pandemic has spread to the whole world, causing over 1.5 million deaths to date”.

Material and methods leave behind unanswered questions.

Results are not clearly structured and the table is way too long and confusing

In the discussion you go more into the background of your study (which could be partly mentioned in your introduction). You advice a new prediction tool for in hospital mortality. However you did a validation in the same paper without background information of the independent patient sample. ???? that is very much questionable for me….

Major issues:

1. what is the Definition of moderate to Severe COVID-19 Patients? Following your suggestion, we have included this information in the manuscript.

2. Lines 221 A total of 2879 moderate to severe COVID-19 hospitalized patients were initially

lines 222 evaluated for inclusion in the development cohort. Of these, 809 were excluded: 515

line 223 remained hospitalized at the time of analysis and 294 were on chronic anticoagulant

line 224 treatment before hospitalization. The final sample consisted of 2070 patients (884

line 225 females and 1186 males) with definite outcomes: 1677 (81.01%) patients had been

line 226 discharged (survivors) and 393 (18.99%) patients had died (non-survivors).

Exclusion of a large group of patients. Approximately 28,1 % of patient were excluded.

A total of 809 patients (approximately 28.1%) were excluded. The main outcome of the study (death/discharge) could not be evaluated in the 515 patients who were still hospitalized at the time of analysis. Thus, their exclusion was mandatory. Additionally, it is reasonable to exclude the 294 patients on chronic anticoagulant treatment before hospitalization. Otherwise, coagulopathy, which is a common complication of severe COVID-19 patients, could have been underestimated.

Follow up time is not clear. (or time to mortality.)? Since the aim of the study was to evaluate in-hospital mortality, the definition of follow up time is the number of days the patient has been hospitalized. As stated in the manuscript, “no data from possible hospital readmission of survivors were available, and it is possible that some initially recovered patients may have worsened a few days later”.

3. it is not clear how many patients were mechanical ventilated?

Due to the massive number of infected subjects, both the Emergency department and the Ward of our institution and of many other hospitals were overwhelmed. The access to invasive ventilation and/or intensive care units were limited or prioritized to young patients developing severe hypoxemic respiratory failure. Therefore, a lot of patients candidate to mechanical ventilation did not get it. That is the reason why we did not analyze it, since the results would have been biased. 

4.lines 262- 264 " They were then examined in a multivariate logistic regression model including 1270 patients to identify independent prognostic factors of moderate/severe COVID-19 in-hospital mortality (Table 2)."

Not clear. Sub-analysis of 2070 patients? Why are the 800 patients excluded from this Analysis. As it is stated in the manuscript (Methods section; Statistical analysis): “Statistically significant variables in the categorical analysis were included in a logistic regression model, performed in a randomly selected training cohort including around 60% of the total amount of patients”.

Is that the independent sample of patients about which we read in the abstract? Not clear. Those 1270 patients are the development cohort (around 60% of the total population).

IF Yes: no characteristics of this patient population present.

Univariable analysis was carried out in the whole cohort, and their characteristics are summarized in Table 1.

5. Line 283 A cut-off of 0.076 was arbitrarily selected.

Please elaborate more why this cut off value and not another? This arbitrary cut-off was selected in order to maximize sensitivity and negative predictive value. In this way, patients classified as low-risk on admission could get either discharged early or derived to other centers without intensive care units with the certainty that their likelihood of dying is not as high as those classified as high risk.

6. If a major Revision is possible, the analysis must be checked by a statistician. One of the authors, who performed the statistical analysis, has a master in biomedical research and statistical analysis of public health data from Universidad Autónoma of Barcelona.

Minor issues

1. The language need to be re-edited, there are many typos as well as weak sentences: lines 85-86 To evaluate the risk of demographic data, medical history, and on-admission laboratory parameters in hospitalized COVID-19 patients to predict mortality

(Not well written)

We have modified this sentence.

Lines 133-135 Clinicians need better predictors of which patients are prone to deteriorate rapidly or who may go on to die

(“predict Mortality” better) not go die!!!!

We have modified this sentence.

I hope now you consider the manuscript acceptable for publication.

Yours sincerely,

Juan Manuel Alonso-Domínguez

---

## [Editor Report · Decision Letter 1]

11 Feb 2021

Development and validation of a predictive model of in-hospital mortality in COVID-19 patients

PONE-D-20-35733R1

Dear Dr. Alonso-Dominguez,

We’re pleased to inform you that your manuscript has been judged scientifically suitable for publication and will be formally accepted for publication once it meets all outstanding technical requirements.

Kind regards,

Aleksandar R. Zivkovic

Academic Editor

PLOS ONE

---

## [Editor Report · Acceptance letter]

23 Feb 2021

PONE-D-20-35733R1 

Development and validation of a predictive model of in-hospital mortality in COVID-19 patients 

Dear Dr. Alonso-Dominguez:

I'm pleased to inform you that your manuscript has been deemed suitable for publication in PLOS ONE. Congratulations! Your manuscript is now with our production department. 

Kind regards, 

on behalf of

Dr. Aleksandar R. Zivkovic 

Academic Editor

PLOS ONE